

# Magnitude correlations in a self-similar aftershock rates model of seismicity

Andres F. Zambrano Moreno[1] and Jörn Davidsen[1,2]

[1]Department of Physics and Astronomy, University of Calgary, 2500 University Drive NW Calgary, Alberta T2N 1N4, Canada
[2]Hotchkiss Brain Institute, University of Calgary, 3330 Hospital Dr NW, Calgary, Alberta T2N 4N1, Canada

**Correspondence:** Andres Zambrano (andres.zambranomoren@ucalgary.ca)

**Abstract.** Crucial to the development of earthquake forecasting schemes is the manifestation of spatiotemporal correlations between earthquakes as highlighted, for example, by the notion of aftershocks. Here, we present an analysis of the statistical relation between subsequent magnitudes for a recently proposed self-similar aftershock rates model of seismicity, whose main distinguishing feature is that of interdependence between trigger and triggered events in terms of a time-varying frequency magnitude distribution. By means of a particular statistical measure, we study the level of magnitude correlations under specific types of time conditioning, explain their provenance within the model framework and show that the type of null model chosen in the analysis plays a pivotal role in the type and strength of observed correlations. Specifically, we show that while the variations in the magnitude distribution can give rise to large trivial correlations between subsequent magnitudes, the non-trivial magnitude correlations are rather minimal. Simulations mimicking Southern California show that these non-trivial correlations cannot be observed at the $3\sigma$-level at the current level of completeness. We conclude that only the time variations in the frequency-magnitude distribution might lead to significant improvements in earthquake forecasting.

## 1  Introduction

An outstanding question in earthquake dynamics is how reliably one is able to *predict* or *forecast* earthquakes. Forecasting can be defined as a statement of relative likelihood of specific earthquake(s) to occur as a function of space, time and magnitude windows, and should be contrasted to the concept of prediction, which is a specific statement of whether an earthquake *will* or *will not* occur at a particular place and time with a certain magnitude (Jackson and Kagan, 1999). In recent years, forecasting of earthquakes has seen a major effort on many different fronts (Gerstenberger et al., 2005; Helmstetter et al., 2006; Holliday et al., 2007; Schorlemmer et al., 2010; Woessner et al., 2010; Zechar et al., 2010; Field and Milner, 2018; Moschetti et al., 2018; DeVries et al., 2018), also see (Ogata, 2013; Tiampo and Shcherbakov, 2013; Ogata, 2017; Michael and Werner, 2018) for reviews. A defining characteristic of earthquakes is their clustering in both space and time. By considering various empirical relations one can construct models (the backbone of a multitude of forecasting efforts), most being a special case of the





Hawkes process (Hawkes, 1971), such as the epidemic type aftershock sequence (ETAS) model (Ogata, 1988, 1998), the branching aftershock sequence (BASS) model (Turcotte et al., 2007), or the every earthquake is a precursor according to scale

(EEPAS) (Evison and Rhoades, 2004; Rhoades and Evison, 2004), all of which exhibit spatiotemporal clustering. It is through the aforementioned constitutive statistical models that forecasting of seismicity is often implemented (Helmstetter et al., 2006; Schorlemmer et al., 2010; Woessner et al., 2010; Zechar et al., 2010). Temporal clustering is exemplified by the increased *rate* (number of earthquakes per unit time) of local seimicity after large earthquakes, where triggering of earthquakes by other earthquakes through either static or dynamic stress changes is one of the predominant physical processes occurring over a wide

range of spatio-temporal scales (Moradpour et al., 2014; Hainzl et al., 2014). The empirically derived Omori-Utsu relation (Omori, 1894; Utsu, 1957) is used to encode temporal clustering in many of the aforementioned models of seismicity, *e.g.*, the extensively studied ETAS model employs this relation. Yet, in its original formulation the Omori-Utsu relation is typically not self-similar. By formulating the rate of earthquakes as a self-similar process which gives rise to a generalized Omori-Utsu relation, one finds greater agreement with observed seismic behavior in Southern California (SC) when compared to

the standard non self-similar form (Davidsen and Baiesi, 2016). Such self-similarity provides support for the hypothesis that one can scale up constitutive rules derived from fracture and friction experiments in the lab to tectonic earthquakes (Scholz, 1990; Mogi, 2007). The *self-similar aftershock rates* (SSAR) model proposed in (Davidsen and Baiesi, 2016) is a model of earthquake-earthquake triggering, similar to the ETAS model. The SSAR model contains two distinguishing features; event rates are self-similar with respect to the magnitude difference between the trigger and the triggered event, and the frequency-

magnitude distribution of the triggered events for a single trigger varies in time. In the standard Omori-Utsu relation the number of triggered events of a particular energy does not simply depend on the magnitude difference between trigger and triggered events (*i.e. mother-daughter* events). This relation is not self-similar unless very special conditions are met; conditions which are typically inconsistent with observations (Davidsen and Baiesi, 2016). The *ansatz* for a self-similar mother-daughter rate relation can be expressed as,

$$r\left(m_{as}, t | m', 0\right) = \frac{1}{\tau_{\Delta m}} f\left(\frac{t}{c_{\Delta m}}\right), \quad (1)$$

where $\tau_{\Delta m}$ and $c_{\Delta m}$ are two time scales that only depend on $\Delta m = m' - m_{as}$, $m'$ and $m_{as}$ being the magnitudes of the trigger and triggered event, respectively, and $t$ is the time interval between the trigger and triggered event. Dependence on the magnitude difference for the two time scales allows one to obtain two scale-free regimes when considering the frequency-magnitude distribution of events triggered by events of the same magnitude; a feature consistent with observed behavior in the

Southern California (SC) catalog (Davidsen and Baiesi, 2016) and shown to exist in the detailed analysis of aftershock rates in Japan (Peng et al., 2007). Furthermore, scaling of the form in Eq. 1 for a particular $f\left(\frac{t}{c_{\Delta m}}\right)$ was shown to be consistent with all accepted empirical relations (Davidsen and Baiesi, 2016). Just as the ETAS model is widely used in forecasting efforts, one would wish to use the SSAR model for this particular purpose given that the latter was shown to better describe the SC seismic data. The main difference between the ETAS and SSAR model is that in the latter case the mother-daughter magnitudes are

effectively coupled in accordance with Eq. 1. From this vantage point, one would like to quantify the strength of the statistical correlation of magnitudes in a time ordered catalog in the SSAR model, which may ultimately aid in developing more reliable





forecasting methods. For this purpose, the magnitude correlations between *subsequent* events are of particular interest. In order to study magnitude correlations between subsequent events we apply here a statistical method similar to the ones employed in (Lippiello et al., 2008; Davidsen and Green, 2011; Davidsen et al., 2012). An important aspect to highlight is that in our

analysis we test two different types of null hypotheses against the SSAR model. We find that the null hypothesis plays a significant role in the types and strength of magnitude correlations observed. This allows us to distinguish between trivial magnitude correlations that are simply a consequence of the variations in frequency-magnitude distribution and non-trivial ones that are not.

## 1.1 Overview of this paper

We first give a brief overview of the SSAR model (Sect. 2), introduce the specifics of the surrogate catalogs (Sect. 2.1), followed

by the methodology (Sect. 3.1) and analysis of the magnitude correlations between subsequent events through the lens of a particular statistical measure (Sect. 3.2, 3.3). In the latter, we show why it is important that in the analysis of magnitude correlations care must be taken with the methodology (on choosing the randomized magnitudes, *i.e.*, the type of null model) if one wishes to avoid confounding factors. In Sect. 4 we present a discussion of our results.

## 2 The self-similar aftershock rates (SSAR) model

The SSAR model recasts the standard Omori-Utsu rate equation into a self-similar version. A distinguishing feature of the rate equation in the SSAR model is that it only depends on the difference between mother-daughter events making it self-similar (*e.g.,* the rates of a magnitude 3 mother and magnitude 2 daughter event are the same as those of a magnitude 5 mother and a magnitude 4 daughter event),

$$r\left(m_{as}, t | m', 0\right) = \frac{1}{\tau_{\Delta m}} \left(\frac{t}{c_{\Delta m}} + 1\right)^{-p}, \tag{2}$$

with time scales,

$$c_{\Delta m} = c_0 10^{g\Delta m} \text{ and } \tau_{\Delta m} = \tau_0 10^{-z\Delta m}, \tag{3}$$

where $g$ and $z$ are universal scaling exponents, with constant prefactors $c_0$ and $\tau_0$, and $p \gtrsim 1$ (Davidsen and Baiesi, 2016) ($p \leq 1$ is unphysical if considering only daughter events *aka* directly triggered events, see, for example, (Davidsen et al., 2015)). To obtain the total number of triggered events of magnitude $m_{as}$ for a given trigger $m'$, we integrate Eq. 2 in the time domain,


$$N\left(m_{as} | m'\right)$$

$$= \int\limits_0^\infty \frac{1}{\tau_{\Delta m}} \left(\frac{t}{c_{\Delta m}} + 1\right)^{-p} dt = \frac{c_{\Delta m}}{\tau_{\Delta m}(p-1)}, \tag{4}$$

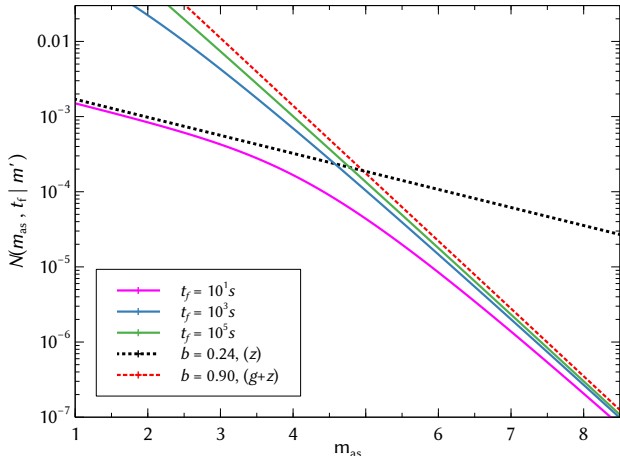

**Figure 1.** Plot of Eq. 7 for $m' = 1.8$ and $t_f = 10^1, 10^3$, and $10^5$ seconds. Dashed lines correspond to the $b$-value exponent; $b = z$ for early times (black) and $b = g + z$ for late times (red).

which only depends on $\Delta m$ ensuring self-similarity. Integrating Eq. 2 from a chosen *cutoff* magnitude $m_{cut}$ to $\infty$, we find

$$N_> \left(m_{cut}|m'\right) = \frac{c_0\, 10^{(g+z)\left(m'-m_{cut}\right)}}{\tau_0\,(p-1)\,(g+z)\ln 10}, \tag{5}$$

which is simply the Gutenberg-Richter relation for triggered events (Davidsen and Baiesi, 2016), giving us the scaling relation,

$$b_{as} = g + z. \tag{6}$$

In contrast, for finite times the number of triggered events of magnitude $m_{as}$ up to a time $t_f$ is,

$$N\left(m_{as}, t_f|m'\right) \equiv \int\limits_0^{t_f} r\left(m_{as}, t|m', 0\right) dt. \tag{7}$$

Plotting Eq. 7 for different values of $t_f$ in Fig. 1, we observe a defining characteristic of the SSAR model. Unlike another self-similar model (Lippiello et al., 2007, 2008), two scale-free regimes co-exist for all finite values of $t_f$ in the frequency-magnitude distribution. Namely, $b \rightarrow z$ corresponds to the dominating exponent in the early time limit of Eq. 2, and a second regime with $b \rightarrow g + z$ that dominates at later times. In other words, at early times we can observe a $b$-value of $z$ over an
extended regime of small magnitudes but as $t_f$ increases the transition point moves towards smaller magnitudes and we begin to see a more extended range with a $b$-value of $g + z$, which corresponds to the asymptotic behavior for $t_f \rightarrow \infty$.

Analogously to the ETAS model, the full SSAR model is given by a time-varying seismic rate (also called the conditional intensity or stochastic intensity), which takes on the following form





$\lambda\left(t, m, \boldsymbol{r}\right)=$

$$\mu\, s_0\left(m\right) + \sum_{t'<t} \kappa\left(m'\right)\psi_{m,m'}\left(t-t'\right)s\left(m\right)\zeta_{m'}\left(\boldsymbol{r}-\boldsymbol{r}'\right) \quad (8)$$

where $\mu$ is a constant and $s_0\left(m\right)$ is the probability density function (PDF) of the magnitudes of background events, i.e., events that are not triggered by other events, and their product determines the background rate, which is assumed to be uniform

in time and space. Similarly, $\psi_{m,m'}\left(t\right), s\left(m\right),$ and $\zeta_{m'}\left(\boldsymbol{r}\right)$ are the PDFs for the temporal distance, magnitude and spatial distance of daughter events triggered by a mother of magnitude $m'$, respectively. $\kappa\left(m'\right)$ corresponds to the total number of daughters triggered by a mother, often denoted as the productivity relation. For the purpose of our temporal analysis of magnitudes, we can ignore the spatial component in Eq. 2, we refer the reader to (Moradpour et al., 2014; Davidsen and Baiesi, 2016) for a treatment of $\zeta_{m'}\left(\boldsymbol{r}\right)$. The PDFs for the magnitude distribution of background and triggered events are the

normalized Gutenberg-Richter relations:

$$\begin{cases} s_0\left(m\right) = \beta\, e^{-\beta(m-m_{cut})} \\ s\left(m\right) = \beta_{as} e^{-\beta_{as}(m-m_{cut})} \end{cases}, \qquad (9)$$

where $m_{cut}$ indicates the lower magnitude cut-off, $\beta = b \ln 10$ and $\beta_{as} = b_{as} \ln 10$. The productivity relation and the normalized temporal distribution for the generalized Omori-Utsu relation are, respectively,

$$\kappa\left(m'\right) = A\, e^{\beta_{as}\left(m'-m_{cut}\right)}, \qquad (10)$$


$$\psi_{m,m'}\left(t\right) = \frac{(p-1)c_{\triangle m}^{p-1}}{(t+c_{\triangle m})^p}. \qquad (11)$$

### 2.1   SSAR model catalogs

Since the SSAR model was tested using a catalog from Southern California (Davidsen and Baiesi, 2016), we focus here on synthetic model catalogs that are comparable to that from Southern California (SC). We would like to point out again that for

the purpose of our analysis of magnitude correlations below, the spatial location of events is not relevant. A seismic catalog generated through the SSAR model consists of both independent background events and its associated $N-th$ generation aftershocks. The catalog is generated by first seeding background events with magnitudes selected from the corresponding frequency-magnitude distribution of Poissonian times, triggered events are then created from the statistical distributions of magnitude and time in an iterative manner (see Sect.S2 in supplemental material of (Davidsen and Baiesi, 2016) for details). A

realization of the SSAR model, which resembles the SC catalog, henceforth referred to as SSAR-SC, was used for the analysis presented in this paper, see Fig. 2. Yet, none of our findings depends on the specific realization as we tested explicitly. The


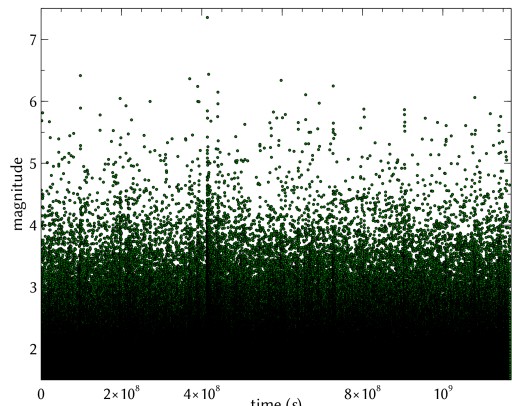

**Figure 2.** Time-magnitude plot for a realization of the SSAR model with a lower magnitude threshold of $1.50$ and $376,311$ events.

SSAR-SC catalog contains $\sim 1.5 \times 10^5$ background events with a total of 376,311 events (after removing $1\%$ of the initial events to minimize boundary effects), lower magnitude cutoff $m_{min} = 1.50$, with model parameters $p = 1.15$, $c_0 = 210\,s$, $\tau_0 = 10^4\,s$, $g = 0.66$, $z = 0.24$ that agree with those of SC (Davidsen and Baiesi, 2016). This corresponds to a coverage of about 36 years,

the current length of the relocated SC catalog from 1981 to 2017 (Hauksson et al., 2012). We have also imposed a hard upper cutoff, $M_{max} = 7.40$, for the largest magnitude possible in the SSAR-SC catalog in order to avoid an unphysical runway process.

## 3    Magnitude correlations in the SSAR model

In this section we aim to answer the question of what is the type and strenght of the effective magnitude correlations in the

SSAR model. Correlations arise by means of the rate equation given by Eq. 2 since the timing of the daughters will depend on the magnitude difference between the daughters and mothers as captured by the functional dependencies of $c_{\Delta m}$ and $\tau_{\Delta m}$. We first draw attention to the methodology since this is a crucial aspect in understanding the types of correlations observed in the model.

### 3.1    Methodology

Our study of magnitude correlations, similar to methods used in (Lippiello et al., 2008; Davidsen and Green, 2011; Davidsen et al., 2012), considers subsequent events in the time ordered catalog. Specifically, it focuses on $\Delta m_i = m_{i+1} - m_i$ (for a particular magnitude thresholds $m_{th}$) and compares these to randomized magnitude differences averaged over 500 realizations, i.e., $\Delta m_i^* = m_{i^*} - m_i$ where $m_{i^*}$ is a magnitude chosen at random. If magnitude correlations between subsequent events in the ordered catalog are present, the distribution of $\Delta m$ will deviate from the distribution of the randomized case; $\Delta m^*$. To asses

whether magnitude correlations exist in the SSAR model we considered three types of conditioning (*unconditioned*, $\Delta t$ and $\Delta t$ & M-D) for various magnitude thresholds $m_{th}$. For the *unconditioned* case we use the quantity $\delta P(m_0) \equiv P(\Delta m < m_0) -$





$P\left(\Delta m^* < m_0\right)$, where $P\left(...\right)$ refers to the cumulative distribution function (CDF) of the ordered and randomized catalogs. For $\Delta t$ and $\Delta t$ & M-D (mother-daughter) conditioning we consider the corresponding quantities,

$\delta P\left(m_0 | \Delta t < y\right)$

$$= P\left(\Delta m < m_0 | \Delta t < y\right) - P\left(\Delta m^* < m_0 | \Delta t < y\right), \quad (12)$$

and

$\delta P\left(m_0 | \Delta t < y \text{ \& M-D}\right)$

$= P\left(\Delta m < m_0 | \Delta t < y \text{ \& M-D}\right)$

$$- P\left(\Delta m^* < m_0 | \Delta t < y \text{ \& M-D}\right). \quad (13)$$

Specifically, for $\Delta t$ and $\Delta t$ & M-D conditioning one only considers subsequent event pairs and their $\Delta m_i$ if the time interval between the two events is not longer than $\Delta t$. In addition, for $\Delta t$ & M-D conditioning these event pairs also have to be a mother-daughter pair. The reason why we choose to condition on time intervals is motivated by the expectation that event

pairs that are closer in time are more likely to be related — either by being mother-daughter pairs or by being daughters of the same mother — than those further apart. Note that in the SSAR model all dependencies are fundamentally encoded at the mother-daughter level, *viz.* Eq. 2. By conditioning on time we are also preferentially "picking" certain magnitude differences via the rate equation, Eq. 2.

Another important aspect in our analysis is how we randomly choose the magnitudes $m_{i*}$ in the case of $\Delta t$ or $\Delta t$ & M-D

conditioning. One can either pick $m_{i*}$'s from the already conditioned catalog — which we call *sub-catalog* randomizing — or one can pick $m_{i*}$'s from the full unconditioned catalog — called *full-catalog* randomizing. It is important to point out that in sub-catalog randomizing the frequency-magnitude distributions of $m_{i+1}$ and $m_{i*}$ are *identical* by construction. In full-catalog randomizing, however, the $m_{i*}$ might or might not follow a different frequency-magnitude distribution. This is possible since the frequency-magnitude distribution can vary in the SSAR model as discussed above.

For all three types of conditioning one can state the following. If the quantity $\delta P\left(m_0 | \ldots\right)$ significantly deviates from $0$ for at least some values of $m_0$, then correlations between subsequent magnitudes are present. The two randomizing methods used in our analysis, one which keeps the frequency-magnitude distribution fixed while the other one might not, both produce in principle different types of magnitude correlations: When the frequency-magnitude distribution is fixed, we are seeing inherent (*non-trivial*) magnitude correlations, while the correlations in the other case correspond to a mixing of non-trivial and trivial

correlations, where the latter simply arise due to the differences in the frequency-magnitude distribution.

### 3.2 Magnitude correlations for the unconditioned case of the SSAR model

In Fig. 3 we show the previously described measure of magnitude correlations for subsequent events for the unconditioned case in the SSAR-SC catalog. Magnitude correlations that are significant at the $3\sigma$-level exist in the SSAR-SC catalog in the

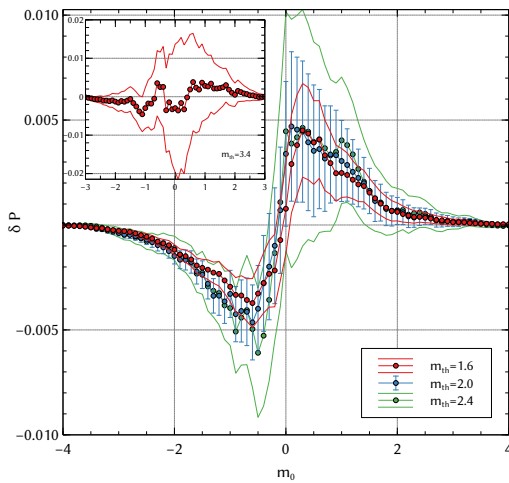

**Figure 3.** Magnitude correlations of the unconditioned SSAR-SC catalog for $m_{th} = 1.60$, $2.00$, $2.40$ and $3.40$ (inset). Error bars correspond to $3\sigma$.

range $m_{th} = 1.60 - 2.80$. Inspecting the slope of $\delta P(m_0)$ in Fig. 3, we see that the values $\Delta m$ have a higher tendency to lie in
a given range when compared to the randomized case $\Delta m^*$, and less likely to lie outside said range: for $m_{th} < 2.4$ the slope
of $\delta P$ is typically positive in the range -0.5 to 0.25 showing an $\approx 0.9\%$ higher probability that $\Delta m$ lies within this range when
compared to $\Delta m^*$. While significant, this difference is very small. The absence of significant correlation at the $3\sigma$-level for
$m_{th} = 3.40$ (the current magnitude of completeness for Southern California (Schorlemmer and Woessner, 2008)) is simply a
consequence of an insufficient number of events.

## 3.3   $\Delta t$ & mother-Daughter conditioning in the SSAR model

To test whether magnitude correlation become stronger if one considers pairs of events that are related, we now focus on the
magnitude correlation analysis for $\Delta t$ & M-D conditioning. The correlations that arise under $\Delta t$ & M-D conditioning are
shown in Fig. 4. The two randomization types produce vastly different results. For sub-catalog randomizing, Fig. 4 a), we see
no qualitative difference in the shape of $\delta P(m_0 | \Delta t < y$ & M-D$)$ compared to the unconditional case. Yet, the probability to
encounter a magnitude difference in an interval around 0, i.e, a daughter event being similar in magnitude to the mother event,
is now up to four times higher for the smallest $\Delta t$ than for $\delta P(m_0)$ (Fig. 3). Even for $\delta P(m_0 | \Delta t < \infty$ & M-D$)$ there is a
significant increase in the excess of daughters that are on average the same size as the mother to about $2.6\%$. As before, this
excess comes at the expense of significantly smaller and larger daughter events. As Fig. 4 c) shows, a similar behavior occurs
for larger $m_{th}$.

40       In contrast, when estimating $\delta P(m_0 | \Delta t < y$ & M-D$)$ using full-catalog randomizing, Fig. 4 b), the distribution is qualita-
tively and quantitatively distinct compared to $\delta P(m_0)$. Specifically, Fig. 4 b) demonstrates that independent of the $\Delta t$ values,
the magnitudes of the daughter events tend to be on average larger than those of the mother events compared to what is expected

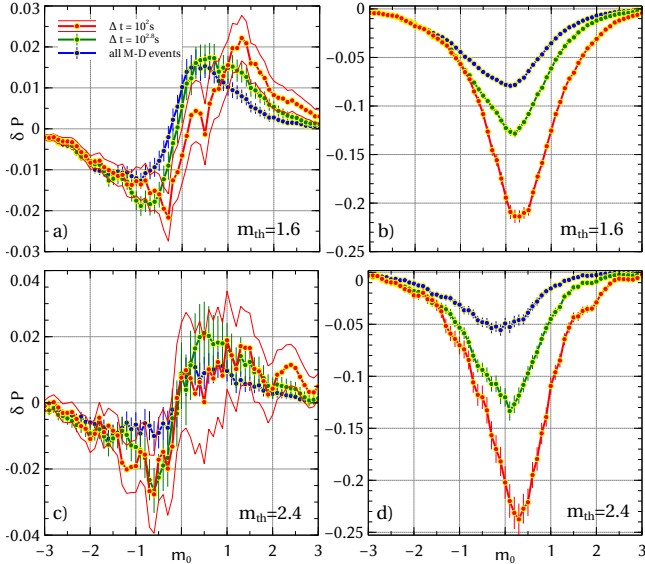

**Figure 4.** Magnitude correlations for the SSAR-SC catalog with $\Delta t$ & M-D conditioning for sub-catalog randomizing (left panels) and full-catalog randomizing (right panels) for $m_{th} = 1.60, 2.40$. To see the trends clearly only $1\sigma$ error bars are shown.

based on the null model. The associated probability increases from a minimum of $8\%$ to over $20\%$ for increasing values of $\Delta t$. As Fig. 4 d) shows, a similar behavior occurs for larger $m_{th}$.

As discussed above, the underlying difference between sub-catalog and full-catalog randomizing for $\Delta t$ & M-D conditioning is the frequency-magnitude distribution of the randomized daughter events (*c.f.,* end of Sect.3.1). Thus, our observations indicate that the non-trivial magnitude correlations as captured by sub-catalog randomizing are significant but smaller by up to a factor of 6 for $\Delta = 10^2$s and $m_{th} = 2.4$ compared to the mixture of trivial and non-trivial magnitude correlations measured by using full-catalog randomizing. This indicates that the trivial magnitude correlations arising from differences in the

frequency-magnitude distribution significantly outweigh the non-trivial ones under appropriate conditioning and play the more dominant role.

While in our model simulations we can readily identify mother-daughter pairs, i.e., the ground truth is known, this is not the case for field data. Thus, for such catalogs — including the SC catalog — one would need to infer mother-daughter pairs *aka* decluster the catalog first (Baiesi and Paczuski, 2004; Zaliapin et al., 2008; Marsan and Lengliné, 2008; Zaliapin and

Ben-Zion, 2013; Gu et al., 2013) in order to estimate the magnitude correlations for $\Delta t$ & M-D conditioning. As an alternative, time conditioning alone has been used in the past (Lippiello et al., 2008; Davidsen and Green, 2011; Davidsen et al., 2012). This is what is shown in Fig. 5. When comparing to Fig. 4, a significant decrease in amplitude of $\delta P$ can be observed. This decrease is especially large for small $m_{th}$. Yet, as before the trivial magnitude correlations arising from differences in the frequency-magnitude distribution significantly outweigh the non-trivial ones.


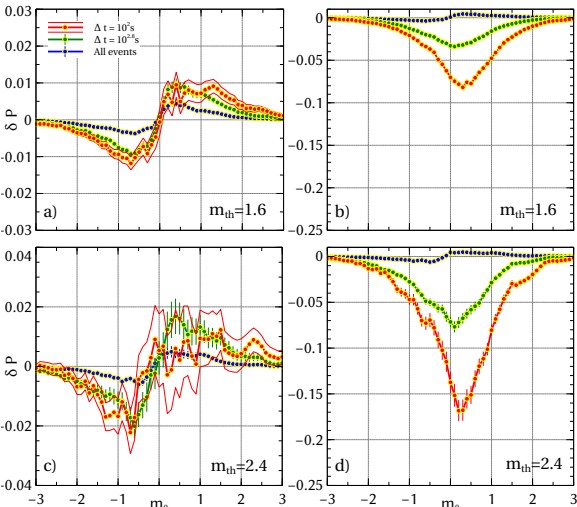

**Figure 5.** Magnitude correlations for the SSAR-SC catalog with $\Delta t$ conditioning for sub-catalog randomizing (left panels) and full-catalog randomizing (right panels) for $m_{th} = 1.60, 2.40$. To see the trends clearly only $1\sigma$ error bars are shown.

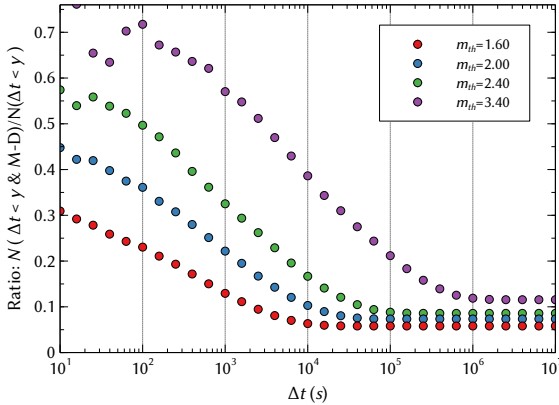

**Figure 6.** Ratio of total number of events satisfying $\Delta t < y$ & M-D to $\Delta t < y$ conditioning for the SSAR-SC catalog for different values of $m_{th}$.

To clarify the reason for the difference between the $\Delta t$ & M-D conditioning and the $\Delta t$ conditioning, we can examine the ratio of event pairs within $\Delta t$ that are mother-daughter to all event pairs that fall within the time interval $\Delta t$ in the SSAR model (Fig. 6). According to Fig. 6, in order to maximize the ratio of mother-daughter events one should choose subsequent event pairs that have a high $m_{th}$ and are close in time (small $\Delta t$). This explains the differences between two different types of conditioning shown in Fig. 4 and in Fig. 5, which become less pronounced for higher $m_{th}$ and smaller $\Delta t$.

The aforementioned maximization comes with a trade-off; although a higher $m_{th}$ value captures more mother-daughter pairs, the total number of selected event pairs goes down at the same time leading to higher statistical uncertainties. It is also


important to realize that the ratios shown in Fig. 6 are for the specific parameters used in our realization of the SSAR model (*cf.* Sect.2.1). Choosing different parameter values for $c_0$ and $\tau_0$, for example, will effect the specific ratio even if one keeps $c_0/\tau_0$ constant, though the qualitatively behavior remains the same.

## 4   Discussion & Conclusion

Through a particular statistical measure (Sect. 3.1) we have shown how two different types of magnitude correlations between

*subsequent* events arise in the SSAR model (Sect. 3.3). Trivial correlations are largely a consequence of variations in the frequency-magnitude distribution, while this is not the case for non-trivial correlations, similar to what has been discussed in the context of tectonic seismicity (Davidsen and Green, 2011). Both types of correlations can be estimated by using different underlying null models, implemented here by the two different types of catalog randomizing. Given that magnitudes in the SSAR model are not independent (as exemplified in Eq. 2), it does not come as a surprise that non-zero magnitude correlations

exist. We were able to explicitly show that it is indeed the mother-daughter pairs that are largely responsible for these correlations. Based on this, we were also able to show that one can increase the observed magnitude correlations by conditioning on shorter time intervals and considering higher values of $m_{th}$ (*c.f.*, Fig. 6); an important fact one can use when triggering relations or declustering algorithms are unknown and only information on time intervals are available, such as in the case of real world catalogs. When dealing with real-world catalogs one needs, however, to consider the effects due to magnitude of

completeness and short term aftershock incompleteness as well (Kagan, 2004; Moradpour et al., 2014; Hainzl, 2016).

Finally, the significantly higher strength of the trivial correlations compared to the trivial correlations is the main outcome of our analysis. Thus, when it comes to improving earthquake forecasting efforts our analysis leads us to believe that looking at the time variations in the frequency-magnitude distribution could perhaps be a more fruitful approach then focusing on non-trivial correlations. Using the SSAR model instead of the ETAS model in existing forecasting frameworks would be one way

to utilize this. This remains a challenge for the future.

*Code and data availability.* Python code, synthetic catalog data and plot data can be found in (Zambrano Moreno, 2019). Code and data are under the the GPLv3 license (GNU.org, 2018). Data for Southern California was downloaded from (Hauksson et al., 2017). For the methodology used on the SC catalog see (Hauksson et al., 2012).

*Competing interests.* AZ and JD have no conflict of interest.

*Acknowledgements.* AZ would like to thank Jordi Baro for for providing an ETAS C++ code which helped greatly in understanding the type of coding that would be required for the creation of the SSAR code and for helpful consultations, to Mohammed Yaghoobi for helpful





discussions on the interpretation of the magnitude correlation plots and to Ayush Mandawal for lending an ear and participating in general dialogues on the topic. JD was financially supported by NSERC.

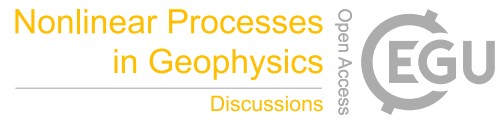

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
