# Peer review of "Magnitude correlations in a self-similar aftershock rates model of seismicity"

_Nonlinear Processes in Geophysics, 2019_

## Referee Comment (RC1) · Anonymous Referee #1 · 10 Aug 2019

article graphicx

**Review of the manuscript *Magnitude correlations in a self-similar aftershock rates model of seismicity* by**

Andres F. Zambrano Moreno and Jörn Davidsen

August 10, 2019

**1   General comments**

The authors test the existence of magnitude correlations for a self-similar earthquake occurrence rate model. As a first observation I would like to remark that magnitude correlations are intrinsic to this kind of models simply because the occurrence probability cannot be factorized.

A second crucial observation is that this kind of model was firstly introduced by (; ; ; ; ; ) and all these articles should be quoted.

[Figure]

**2 Specific comments**

As stated in the previous section this approach is not new. The only difference is in the introduction of two scaling exponents instead of only one. More precisely the previous model used an occurrence rate model described by

$$r(m_{as}, t|m', 0) = f\left(\frac{t}{c_{\Delta m}}\right) \tag{1}$$

with $c_{\Delta m} = c_0 10^{b\Delta m}$

Conversely the new self-similar model uses

$$r(m_{as}, t|m', 0) = \frac{1}{\tau_{\Delta m}} f\left(\frac{t}{c_{\Delta m}}\right) \tag{2}$$

with $\tau_{\Delta m} = \tau_0 10^{g\Delta m}$ and $c_{\Delta m} = c_0 10^{z\Delta m}$ where $b = g + z$

The authors should discuss the advantage of introducing the two exponents in respect of using only one.

The only novelty in the article is represented by the introduction of the sub-catalog randomizing. This aspect remain, however, obscure and should be better described and discussed. In particular, at my opinion, the differences between the sub-catalog randomizing and the full-catalog randomizing are not sufficiently enlightened. Moreover I suggest that the sub-catalog randomizing should be applied to real catalogs.

**References**

Eugenio Lippiello, Cataldo Godano, and Lucilla de Arcangelis. Dynamical scaling in branching models for seismicity. *Phys. Rev. Lett.*, 98:098501, Feb 2007.

E. Lippiello, M. Bottiglieri, C. Godano, and L. de Arcangelis. Dynamical scaling and generalized omori law. *Geophysical Research Letters*, 34(23):L23301, 2007.

E. Lippiello, L. de Arcangelis, and C. Godano. Time, space and magnitude correlations in earthquake occurrence. *International Journal of Modern Physics B*, 23(28n29):5583–5596, 2009.

E. Lippiello, C. Godano, and L. de Arcangelis. The earthquake magnitude is influenced by previous seismicity. *Geophysical Research Letters*, 39(5):L05309, 2012.

E. Lippiello, C. Godano, and L. de Arcangelis. Magnitude correlations in the olami-feder-christensen model. *EPL (Europhysics Letters)*, 102(5):59002, 2013.

Lucilla de Arcangelis, Cataldo Godano, Jean Robert Grasso, and Eugenio Lippiello. Statistical physics approach to earthquake occurrence and forecasting. *Physics Reports*, 628:1 – 91, 2016. Statistical physics approach to earthquake occurrence and forecasting.

---

## Referee Comment (RC2) · Robert Shcherbakov (Referee) · 16 Aug 2019

The paper presents an analysis of the magnitude correlations in the earlier proposed self-similar aftershock rates (SSAR) model by Davidsen and Baiesi,(2016). The paper is well structured and written. It provides a detailed and rigorous analysis of magnitude correlations in the SSAR model. I think it represents an important contribution to the studies of marked point processes. Therefore, I think the paper should be considered for publication in Nonlinear Processes in Geophysics provided the authors address several my concerns.

- I think the authors should mention and cite two references, where the scaling for

aftershock rates was recognised earlier than what is given in Davidsen and Baiesi (2016). Specifically this was done in the following works:

1. R. Shcherbakov, D.L. Turcotte, and J.B. Rundle, "A generalized Omori's law for earthquake aftershock decay", Geophys. Res. Lett., 31 (2004) L11613, doi:10.1029/2004GL019808.

2. R. Shcherbakov, D.L. Turcotte, and J.B. Rundle, "Complexity and Earthquakes" in Treatise on Geophysics, 2nd ed., Vol. 4, Ch. 24, ed. H. Kanamori, Elsevier, 2015, doi:10.1016/B978-0-444-53802-4.00094-4.

Specifically, in Shcherbakov et al. (2015) the scaling relationships given in Eq. (3) was introduced and the corresponding relationship between exponents was also given. In addition, the variability of the b-value during different stages of aftershock sequences was also recognised in Shcherbakov et al. (2004).

- Page 5. Lines 30-35. I think the reference to the Southern California catalogue to explain the absence of correlations in synthetic data is not clear. The synthetic catalogue should not be affected by the incompleteness issue assuming that the simulations were performed properly. So this indicates that correlations diminish with increasing lower magnitude cutoff $m_{th}$. Any explanation for this effect?

- Page 4 Lines 15-24. What are the $\beta$ and $\beta_{as}$ values used in the simulations to generate the synthetic earthquake catalogue? How the difference between these two values affects the "trivial correlations" between magnitudes?

Robert Shcherbakov

---

## Author Comment (AC1) · 22 Oct 2019

[book]book amsthm

ROBERT SHCHERBAKOV COMMENTS
RC: referee's comment, AC: authors' reply to comment

RC1: "I think the authors should mention and cite two references, where the scaling for aftershock rates was recognised earlier than what is given in Davidsen and Baiesi (2016). Specifically this was done in the following works:

1. R. Shcherbakov, D.L. Turcotte, and J.B. Rundle, "A generalized Omori's law for earthquake aftershock decay", Geophys. Res. Lett., 31 (2004) L11613, doi:10.1029/2004GL019808.

2. R. Shcherbakov, D.L. Turcotte, and J.B. Rundle, "Complexity and Earthquakes" in Treatise on Geophysics, 2nd ed., Vol. 4, Ch. 24, ed. H. Kanamori, Elsevier, 2015, doi:10.1016/B978-0-444-53802-4.00094-4."

AC1: Both references have been added to the revised manuscript. The first one is referenced after Eq.(7) and the second reference was added after Eq.(3).

RC2: "- Page 5. Lines 30-35. I think the reference to the Southern California catalogue to explain the absence of correlations in synthetic data is not clear. The synthetic catalogue should not be affected by the incompleteness issue assuming that the simulations were performed properly. So this indicates that correlations diminish with increasing lower magnitude cutoff $m_{th}$. Any explanation for this effect?"

AC2: We believe that this refers to our discussion of Fig. 3 (page 7-8, section 3.2). There the absence of significant correlations for large $m_{th}$ is simply a consequence of a lack of statistics (given the finite duration of the catalog). This was stated at the end of page 8. Please note that the synthetic catalogs do not suffer from incompleteness.

RC3: "- Page 4 Lines 15-24. What are the $\beta$ and $\beta_{as}$ values used in the simulations to generate the synthetic earthquake catalogue?"

AC3: They are $\beta = 2.49$ and $\beta_{as} = 2.07$. The corresponding values for $b$ and $b_{as}$ are now explicitly given in section 2.1.

RC4: "How the difference between these two values" — $\beta_{as}$ and $\beta$ — "affects the "trivial correlations" between magnitudes?"

[Figure]

AC4: Varying the values of $b$ and $b_{as}$ over relevant ranges ($0.5 - 1.6$), where we also allow variations in $z$ and $g$ individually, does not change our findings qualitatively. The larger the difference between $\beta$ and $\beta_{as}$, the more pronounced the trivial correlations become. At the same time the non-trivial correlations remain largely unaltered. We have added a corresponding discussion to the end of section 3.

---

## Author Comment (AC2) · 22 Oct 2019

[book]book amsthm

ANONYMOUS REFEREE #1 COMMENTS

RC1: "The authors test the existence of magnitude correlations for a self-similar earthquake occurrence rate model. As a first observation I would like to remark that magnitude cor- relations are intrinsic to this kind of models simply because the occurrence probability cannot be factorized."

AC1: This is correct as we state in our paper. Yet, it is necessary to quantify the strength and type of the correlations. This is what our paper does.

RC2: "A second crucial observation is that this kind of model was firstly introduced by (; ; ; ; ; ) and all these articles should be quoted."

AC2: We had already given the two main references to this previous work (Lippiello, E., Godano, C., and de Arcangelis, L.: Dynamical Scaling in Branching Models for Seismicity, Physical Review Letters, 98, 098 501, 2007; Lippiello, E., de Arcangelis, L., and Godano, C.: Influence of Time and Space Correlations on Earthquake Magnitude, Physical Review Letters, 100, 038 501, 2008.). In particular, we had already discussed the difference between the model proposed in these papers and the SSAR model just after Eq.(7). In brief, that model is a special case of SSAR corresponding to z=0 and it predicts a logarithmic divergence of the Omori-Utsu rate at short times (Lippiello, E., Bottiglieri, 5 M., Godano, C., and de Arcangelis, L.: Dynamical Scaling and Generalized Omori Law, Geophysical Research Letters, 34, 2007a.). However, both features are not consistent with high-resolution earthquake catalogs from Southern California (Davidsen, J. and Baiesi, M.: Self-Similar Aftershock Rates, Physical Review E, 94, https://doi.org/10.1103/PhysRevE.94.022314, 2016.). We have now added the Lippiello et al. (2007a) reference and expanded our discussion of their model throughout our paper.

RC3: "As stated in the previous section this approach is not new. The only difference is in the introduction of two scaling exponents instead of only one... The authors should discuss the advantage of introducing the two exponents in respect of using only one."

AC3: Having two relevant time scales and, hence, two scaling exponents is a crucial generalization as it matches with observational data from Southern California as already stated in our paper (page 2). See AC2 directly above for a more detailed

response and the revisions we made.

RC4: "The only novelty in the article is represented by the introduction of the sub-catalog randomizing. This aspect remain, however, obscure and should be better described and discussed, In particular, at my opinion, the differences between the sub-catalog randomizing and the full-catalog randomizing are not sufficiently enlightened."

AC4: We disagree with the referee regarding the novelty. Our paper quantifies for the first time the type and strength of the magnitude correlations arising in the SSAR model. The SSAR model is a non-trivial generalization of earlier attempts to provide a self-similar extension of the Omori-Utsu relation and it is consistent with high-resolution earthquake catalogs from Southern California as shown previously. To establish the type of correlations in the SSAR model, we use different null models or randomization procedures. To clarify these procedures, we have now added supplemental material providing a more detailed description and reference it in section 3.1.

RC5: "Moreover I suggest that the sub-catalog randomizing should be applied to real catalogs."

AR5: This has already been done in (Davidsen and Green, 2011). We have added a corresponding remark at the beginning of section 3. It is important to realize, though, that for real-world catalogs one does not have direct access to the triggering relations between events. In addition, one needs to consider, for example, the effects of short-term aftershock incompleteness as well. Thus, a direct comparison is outside the scope of the paper at hand but will be investigated in the future.

---

## Author Response (AR1)

ROBERT SHCHERBAKOV COMMENTS
RC:referees comment, AC:authors comment

RC1: "...the authors should mention and cite two references, where the scaling for aftershock rates was recognised earlier than what is given in Davidsen and Baiesi (2016). Specifically this was done in the following works: 1. R. Shcherbakov, D.L. Turcotte, and J.B. Rundle, "A generalized Omoris law for earthquake aftershock decay", Geophys. Res. Lett., 31 (2004) L11613, doi:10.1029/2004GL019808. 2. R. Shcherbakov, D.L. Turcotte, and J.B. Rundle, "Complexity and Earthquakes" in Treatise on Geophysics, 2nd ed., Vol. 4, Ch. 24, ed. H. Kanamori, Elsevier, 2015, doi:10.1016/B978-0-444-53802-4.00094-4."

AC1: Both references have been added to the revised manuscript. The first one is referenced just before Eq.(6) (also added reference in page 4 (line 17)) and the second reference was added before Eq.(2).

RC2: "- Page 5. Lines 30-35. I think the reference to the Southern California catalogue to ex- plain the absence of correlations in synthetic data is not clear. The synthetic catalogue should not be affected by the incompleteness issue assuming that the simulations were performed properly. So this indicates that correlations diminish with increasing lower magnitude cutoff $m_{th}$. Any explanation for this effect?"

AC2: We believe that this refers to the discussion of Fig. 3 (page 7-8, section 3.2). There the absence of significant correlations for large $m_{th}$ is simply a consequence of a lack of statistics (given the duration of the catalog). This was stated at the end of pag.8 (lines 15-17).

RC3: "- Page 4 Lines 15-24. What are the $\beta$ and $\beta_{as}$ values used in the simulations to generate the synthetic earthquake catalogue?..."

AC3: The values were added to the revised manuscript. In page 3 (line 28) the values used in the simulation for $g$ and $z$ are now stated which are related to $b_{as}$ (Eq.(6)), from which $\beta_{as}$ can be obtained. Page 5 (Line 11) now contains the value of $b$ used in the simulation from which $\beta$ can be obtained.

RC4: "..How the difference between these two values" $-\beta_{as}$ and $\beta-$ "affects the "trivial correlations" between magnitudes?"

AC4: (Discuss fresh plots)

————————————————

ANONYMOUS REF COMMENTS

RC1: "The authors test the existence of magnitude correlations for a self-similar earthquake occurrence rate model. As a first observation I would like to remark that magnitude cor- relations are intrinsic to this kind of models simply

because the occurrence probability cannot be factorized."

AC1: Correct, but it is necessary to quantify the correlations.

RC2: "A second crucial observation is that this kind of model was firstly introduced by (; ; ; ; ; ) and all these articles should be quoted."

AC2: We give the two main references to this previous work and we have discussed the difference of the other model with respect to the SSAR model just after Eq.(7) (also providing reference here to the 2007 and 2008 papers pointed out by the referee).

*Joerns comment:"We can probably add one or two more references and expand the discussion of the differences there and add a sentence around line 34 (page 2) but that's it. In particular, their model does not agree with the data from SC. In particular, their model does not agree with the data from SC."
*Andress comment of the above: Added references to the two 2007 papers in Page 2 (lines 3-5). Not sure how we'd add the 2009-2013 references since these are just an application of their model. The 2016 is just a 'summary' of their work thus far. Should we add that their model does not agree with SC on page 4 (line 16): "Unlike another self-similar model (Lippiello et al., 2007a, 2008)..."?

RC3: "As stated in the previous section this approach is not new. The only difference is in the introduction of two scaling exponents instead of only one... The authors should discuss the advantage of introducing the two exponents in respect of using only one."

AC3: Having a second exponent is fundamentally different. The discussion for the advantage of two exponents was mentioned on page 2 (lines 30-34) (i.e. matches the observational data).

RC4: "The only novelty in the article is represented by the introduction of the sub-catalog randomizing. This aspect remain, however, obscure and should be better described and discussed, In particular, at my opinion, the differences between the sub-catalog randomizing and the full-catalog randomizing are not sufficiently enlightened."

AC4: Add supplemental material expanding on randomization. Refer to this on pages 7 and 8.

RC5: "...Moreover I suggest that the sub-catalog randomizing should be applied to real catalogs."

AR5: For the purpose of our investigation this is outside the scope of this paper. We have added a reference to a paper which does carry out such an analysis on the SC catalog in page 6 (line 18).